# Liquid Biopsy for Monitoring EC Patients: Towards Personalized Treatment

**DOI:** 10.3390/cancers14061405

**Published:** 2022-03-09

**Authors:** Raquel Piñeiro-Pérez, Miguel Abal, Laura Muinelo-Romay

**Affiliations:** 1Translational Medical Oncology Group (Oncomet), Health Research Institute of Santiago de Compostela (IDIS), University Hospital of Santiago de Compostela (SERGAS), Trav. Choupana s/n, 15706 Santiago de Compostela, Spain; raquel.pineiro.perez@rai.usc.es; 2Centro de Investigación Biomédica en Red de Cáncer (CIBERONC), Monforte de Lemos 3-5, 28029 Madrid, Spain

**Keywords:** personalized medicine, endometrial cancer, liquid biopsy, monitoring markers, therapy selection

## Abstract

**Simple Summary:**

Although the field of liquid biopsy is clearly having an effect on other tumour types, in endometrial cancer (EC) there is important work to do to implement the analysis of circulating biomarkers into the clinical routine. One of the most evident contexts of application is the disease follow-up in both localized and advanced diseases, which at present is primarily made by imaging techniques. In the present review, we conducted an overview of the circulating biomarkers with the potential to be used as monitoring biomarkers in endometrial tumours and highlighted the key challenges for their translation into the patients’ management in order to help researchers to better focus their work in this field.

**Abstract:**

Endometrial cancer (EC) is the most frequent gynecological cancer in developed countries and its incidence shows an increasing trend. Fortunately, the prognosis of the disease is good when the tumour is diagnosed in an early phase, but some patients recur after surgery and develop distant metastasis. The therapy options for EC for advanced disease are more limited than for other tumours. Therefore, the application of non-invasive strategies to anticipate the recurrence of localized tumours and guide the treatment in advanced stages represents a clear requirement to improve the survival and quality of life of patients with EC. To achieve this desired precision oncology, it is necessary to invest in the identification and validation of circulating markers that allow a more effective stratification and monitoring of patients. We here review the main advances made for the evaluation of circulating tumour DNA (ctDNA), circulating tumour cells (CTCs), circulating extracellular vesicles (cEVs), and other non-invasive biomarkers as a monitoring tool in the context of localized and advanced endometrial tumours, with the aim of providing a global perspective of the achievements and the key areas in which the use of these markers can be developed into a real clinical tool.

## 1. Introduction

Endometrial cancer (EC) is one of the most common malignancies in the developed world, and its incidence has been increasing during over last decade [1]. Up to 142,000 cases are diagnosed worldwide every year [2], and its mortality rate exceeds 2.3/100,000 women [3]. The incidence of this gynecological disease increases with age, with the mean age at diagnosis being 61 ± 2 years, and 90% of cases occur after the age of 50 years [4]. Although most patients with the disease have a good prognosis due to early diagnosis, around 15–20% of these tumours exhibit an aggressive phenotype [5]. Worryingly, the death rate for EC has been increasing during the past 20 years, rising by 8% over the last 13 years [4,6,7,8].

Traditionally this tumour has been classified according to tumour histopathology in two types of endometrial tumours [9]. Type I, also known as estrogen-related endometrioid carcinomas (EEC), represents up to 70% of EC cases and is characterized by low-grade tumours with a good prognosis. Type II tumours, also known as non-endometrioid endometrial cancer (NEEC), are not associated with estrogen regulation and are usually serous and clear cell carcinomas [10], which normally show a worse prognosis [9,11,12,13]. More recently, a molecular classification for these tumours has also been created. According to The Cancer Genome Atlas (TGCA), EC is classified into four different groups based on the presence of somatic mutations, copy number alterations and microsatellite instability status [14]. The first group, known as *POLE* ultramutated, includes tumours with inactivating mutations in POLE exonuclease. MSI hypermutated corresponds to the hypermutated/microsatellite unstable (MSI) group, while the copy-number low (CN low) corresponds to those tumours with low copy-number aberrations, and copy-number high (CN-high) to those with high copy-number alterations [15]. The last group is mainly composed of serous tumours, normally characterized by *p53* alterations and a poor outcome [16].

In EC, total hysterectomy and bilateral salpingo-oophorectomy are usually per-formed as primary treatment. In a few cases, omentectomy and retroperitoneal lymph-node dissection are also applied [2]. Other treatments such as radiotherapy are used for treating pelvic lymph-node regions with the risk of having microscopic malignant lesions [2], such as vaginal brachytherapy. In metastatic or recurrent EC, chemotherapy is the most common treatment [17], the standard first-line therapy being the combination of carboplatin and paclitaxel. Nevertheless, although recent efforts have centered on adding other agents of interest to this standard therapeutic regimen such as metformin, temsirolimus, and bevacizumab, the response rates to them in advanced tumors as first-line therapy do not exceed 40%. Of note, the response rates to second-line chemotherapy, normally based on the use of paclitaxel, have historically been quite poor (<20%) [18]. In addition to the standard treatment based on chemotherapy regimens, hormone treatment and immunotherapy represent the unique targeted therapies that are currently available for the treatment of advanced EC [19].

Molecular studies, such as that performed by the TCGA, have revealed the landscape of genomic alterations present in EC, and have provided valuable insight into the pathogenesis of this disease [20,21]. However, today, molecularly guided management of EC lags behind most other common cancers. For example, for breast or lung tumours, there are a variety of molecular markers to guide the treatment options, which include combinations of targeted therapies for patients with advanced and/or recurrent disease [22,23]. Actually, there are no routine biomarkers applied to monitor endometrial tumours. Taking into account its rising incidence and the limited tools to identify the patients with the worst prognosis and therapeutic response, there is a clear need for the identification of biomarkers to predict recurrence and assess disease response. These markers are essential in applying a personalized treatment that minimizes side effects associated with chemotherapy and radiotherapy and optimizes the selection of those patients who will benefit from targeted drugs and immunotherapy.

A key factor to really personalize the management of EC patients is to complement the tissue-based analyses, which require invasive surgical procedures associated with mortality and patient anxiety, with the analysis of circulating biomarkers. Thus, the present review is directed to provide an overview of different circulating biomarkers described as potential tools to monitor EC patients at the different stages of the disease, from diagnosis to the therapy follow-up of advanced tumours.

## 2. Liquid Biopsy for Personalized Oncology

Current strategies performed to analyse and monitor EC rely on traditional biopsy, an invasive method, which is an essential part of the diagnostic and prognostic workup. Although traditional biopsy has been considered as the gold-standard approach for several years [24], despite its undoubted value, the invasiveness and the lack of representation of the tumour heterogeneity make it an inaccurate tool for disease monitoring [25]. This fact justifies the effort developed by the scientific community to finding new non-invasive methods for longitudinal sampling [10]. Thanks to this effort during the last years, liquid biopsy analyses have emerged as a valuable non-invasive alternative to understanding the molecular characteristics of the tumour in a comprehensive and dynamic way [24]. In contrast to traditional biopsy, liquid biopsy provides real time information of the tumour, allowing for the monitoring of its evolution and its response to therapy [10].

The term “liquid biopsy” was coined by Pantel & Panabiéres in 2010 for the analysis of circulating tumour cells (CTCs) in blood from cancer patients [26]. Today, the term is widely used in oncology to refer to the sampling of tumour-derived material from different liquid biological sources, primarily blood, but also from other body fluids such as saliva, urine, cerebrospinal fluid, ascites, or pleural effusions [27]. Of note, the tumour-derived material present in body fluids includes CTCs, circulating extracellular vesicles (cEVs), circulating tumour DNA (ctDNA), miRNAs and proteins [10]. The analysis of this circulating elements has a great potential for improving many clinical contexts including cancer genotyping at diagnosis, detection of minimal residual disease (MRD) after surgery, monitoring the therapy response, and the appearance of early progressions or resistance to treatment [28].

Currently, liquid biopsy-based tests directed to characterize cell free DNA (cfDNA) are approved as companion diagnostics to select targeted therapies in a reduced number of tumours, such as lung or breast cancer [29,30]. This term refers to DNA fragments released from normal and cancerous cells, being the last named ctDNA. However, this indication is mainly limited to metastatic stages and the number of patients benefitting from these techniques is currently low [31]. Importantly, evidence about the value of liquid biopsy for monitoring patients during treatment is rapidly increasing. For example, different studies have demonstrated that ctDNA monitoring is useful for detecting the presence of MRD after surgery in localized colorectal and other cancer types. Thus, CRC patients with positive post-surgery ctDNA levels have a higher risk to recur than those with undetectable levels [32,33]. In the same line, increments in ctDNA or CTCs levels have been associated with disease progression of metastatic lung and breast cancer patients, even before the confirmation of progression by radiological assessment [34,35,36,37]. Additionally, liquid biopsy assessment has a clear role for monitoring and characterizing the resistance to targeted therapies as many works have demonstrated in the last years in the context of hormone therapy in breast cancer patients and anti-EGFR strategies in mCRC or immunotherapy in NSCLC [38,39,40].

Despite the advances reached in these tumour types for the clinical application of circulating biomarkers to improve the patient management, in gynecological tumours, such as ovarian and endometrial cancer, there is still important work to do in order to implement the use of fluid samples into the clinics [10]. In this sense, the analysis of uterine aspirates as an alternative biopsy sample to diagnose and characterize endometrial tumours has shown advantages in comparison to traditional tissue biopsies. Uterine aspirates consist of endometrial biopsies obtained by aspiration. They are composed of cellular fraction (tumour and stroma cells) and uterine fluid (mainly secretions from the luminal epithelium and glands) [41]. This sample is minimally invasive and reflects molecular alterations present in tissues from the female genital tract. For this reason, uterine aspirates have been described as a sensitive tool for EC diagnosis [42] and also as an accurate strategy to characterize genetic aberrations, taking into account the intra-tumor heterogeneity [43]. Actually, our research group has used uterine aspirate samples to characterize the genetic landscape of EC and develop a personalized approach generating preclinical models for therapy testing and also a monitoring tool through the analysis of cfDNA [44]. In fact, as was previously highlighted, the use of circulating biomarkers in the context of endometrial tumours has a special interest as a monitoring tool for localized and metastatic disease, since traditional follow-up models are not enough to reduce the tumour mortality [45,46]. We here summarize the knowledge about the potential of cfDNA, CTCs and cEVs as a follow-up tool for improving EC management, taking into account two clinically different scenarios, the localized and advanced disease.

## 3. Circulating Biomarkers as a Monitoring Tool to Anticipate Recurrence in Localized EC

The majority of endometrial tumours are diagnosed early, and patients usually have a five-year survival rate of 90%. Unfortunately, up to 20% of the lesions progress to an advanced stage carcinoma, whose 5-year survival rate drops to 15% [47]. Therefore, the development of clinical tools to better stratify the risk of recurrence at surgery and anticipate the potential recurrence before clinical symptoms represents a priority for researchers.

As of today, there are two main serum circulating biomarkers proposed to follow-up the evolution of EC after surgery, which correspond to the human epididymis protein 4 (HE4) and CA125 [48]. Serum levels of both markers were associated with risk factors and described as prognostic tools. Furthermore, different studies have reported the increment of both markers when EC patients recur, whose increment occurs even earlier than clinical confirmation [48,49,50,51]. In particular, Abbink et al. showed a significant association of both biomarkers and clinical characteristics of the tumour, such as high tumor grade, advanced stages, myometrial invasion and lymph node involvement. However, a greater diagnostic capacity has been seen in HE4 when diagnosing a recurrence in the follow-up of the disease [48]. Despite these findings, these biomarkers have not been implemented in clinical practice in the diagnosis and monitoring of EC. Some challenges for their clinical implementation are the use of appropriate thresholds to stratify patients and also the management of physiological factors that can modulate their levels such as age, BMI or renal function [51].

The clearest alternative to serum proteins among other circulating biomarkers is the cfDNA. Although most of the research directed to the study of cfDNA in non-metastatic EC focuses on the search for diagnostic biomarkers, there are a growing number of studies directed to validate cfDNA assessment as a prognostic and monitoring tool. Cicchillitti et al. analyzed cfDNA levels in EC patients and benign controls by qRT-PCR and observed a high correlation between high cfDNA levels and high-risk stages [52]. Recent studies have also shown the prognostic potential of cfDNA fragmentation in other tumour types [53]. In this sense, a study developed in 2018 analyzed cfDNA levels and integrity analyzing by qPCR, Alu115, and Alu247 fragments in serum from a cohort of 60 EC patients, resulting in increased cfDNA values and a lower integrity index in patients with high risk tumours [54]. These markers were associated with lymphovascular invasion (also known as LVSI), another key prognostic factor for disease relapse and poor survival. More specifically, an increase in cfDNA levels was observed in those patients with high-grade endometrial carcinoma with LVSI. At the same time, in an analysis carried out to associate the levels of low molecular weight (LMW) serum cfDNA with various clinical characteristics of the tumor, an association was observed between the concentration of cfDNA and LVSI which supported the association of cfDNA levels with a worse prognosis and overall survival [55]. Other studies also described higher cfDNA levels in high grade and p53 mutated EC tumors, reinforcing their interest as a prognostic factor [44,52]. In addition to cfDNA monitoring, detection of ctDNA at surgery through the identification of a tumour-specific mutation has been associated with conventional risk factors such as high grade, deep myometrial infiltration and FIGO advanced stages [44,56], and also with a worse prognosis [57,58]. Actually, Casas-Arozamena et al. also demonstrated the potential of ctDNA analyses to determine the tumour burden of EC patients that suffered a recurrence after surgery, since the results obtained during the post-surgery follow-up indicated the presence of ctDNA in three patients with progressive disease [44]. More recently, Moss et al. developed a pilot study to decipher the value of ctDNA determination for detecting and monitoring EC recurrence in patients with localized and advanced disease, being able to anticipate the disease recurrence in part of the non-advanced cases, even in those with stage I tumours [59]. Also, in a small cohort of high-risk endometrial tumours (*n* = 9), Feng et al. associated the presence of ctDNA at surgery with late FIGO stages and the presence of lymph node affectation. They monitored ctDNA levels after surgery and found ctDNA in 44% (four of nine) of the patients, showing positive disease relapse in three of these cases [60]. Notably, in this proof-of concept study, ctDNA was superior to HE4 and CA125 markers in predicting disease recurrence.

Although endometrial tumours are not considered as important CTC shedders, the presence of this tumour population has been evaluated by applying different technologies in patients with EC. Ni et al. analyzed the Epithelial Cell Adhesion Molecule (EpCAM) positive CTCs levels in patients with high or intermediate risk endometrial tumours using the CellSearch technology at time of surgery. They found an association between the presence of CTCs and cervical involvement, while no correlation was found between CTCs and serum CA125/HE4. CTC levels did not show an increment after the first cycle of standard chemotherapy, although the monitoring cohort was not enough to generate any robust conclusion about the value of CTCs as a follow-up tool [61]. Our group, as part of ENITEC (European Network for Individualized Treatment in EC), described CTC positivity in the 22% of a cohort of high-risk EC patients, and characterized these CTC with a plasticity phenotype associated with high-risk of recurrence [62]. More recently, we analyzed the CTCs levels in a new cohort of 36 EC patients from which the 38.9% were CTCs positive at surgery, finding higher rates of positivity in high-risk and recurrent disease, although without statistical significance [44]. Unfortunately, despite the evidence generated about the association of CTCs presence and risk factors, there is no robust information about the potential of CTCs as a post-surgery marker.

On the other side, the analysis of circulating EVs represents a promising alternative for improving the detection of protein and genetic tumour biomarkers in gynecologic cancers [63]. However, as it happens with the CTCs field, their value to monitor EC is still unexplored. It is important to keep in mind that EVs are key mediators during tumour development and spread, being recognized as major players in the communication between the tumour and the microenvironment [64,65]. Different plasma EVs-associated proteins have been found to be increased in endometrial tumours and associated with risk factors. Thus, we identified higher levels of ANXA2 in cEVs isolated from plasma samples of patients with EC than in healthy controls. These levels correlated with the risk of recurrence and the tumour histology, suggesting its value as a non-invasive diagnostic and follow-up marker [63]. Plasma exosomes containing LGALS3BP have been described as mediators for progression promoting tumour growth and angiogenesis during EC, and showed potential value for EC diagnosis and prognosis [66]. However, their levels have not been explored in longitudinal samples. In the same line, CLU, ITIH4, SERPINC1, and C1RL levels in exosomes from the serum of EC patients have been reported as accurate diagnostic markers and represent an interesting signature with potential for interrogating the disease evolution [67].

Finally, circulating miRNAs such as miR-100, miR-151a-5p, miRNA205, miR484, miR23a or miR-135b have also been proposed as potential tools for EC diagnosis and monitoring [68,69,70,71,72,73]. MiRNAs are a type of non-coding RNA whose length encompasses 19–25 nucleotides and with an important function blocking translation or degradation of mRNA. This activity is known to be altered in diverse cancers, including EC. Unfortunately, there are few studies that validate circulating miRNAs as monitoring tools in this tumour. One of these studies, addressed by Tsukamoto et al., identified three miRNAs (miR-135b, miR-205 and miR-30a-3p) in the plasma of EEC with diagnostic potential which also showed potential for disease monitoring since they showed decreased levels after hysterectomy [69].

## 4. Circulating Biomarkers as a Monitoring Tool to Guide the Treatment in Advanced EC

Advanced or recurrent EC is not amenable to curative therapies, with the prognosis of patients being quite poor due to the limited treatment options when they progress to chemotherapy. Although new targeted therapies are being assayed in different clinical trials, their current clinical application is in the minority mainly because of the lack of predictive biomarkers that can be interrogated within the disease evolution [19].

Endocrine therapy is considered for the treatment of recurrent/metastatic low-grade EEC with expression of estrogen (ER) and progesterone (PR) receptors, and low disease burden. However, tumours without receptor expression may also respond to this therapeutic strategy [74]. Among the little knowledge we have about the resistance mechanism to the hormone blockage, mutations in ESR1 are gaining interest in EC [75]. On the other hand, the use of the anti-PD-1 monoclonal, Pembrolizumab, has received FDA approval for the agnostic treatment of MMR-deficient or MSI-high tumours, including EC [76]. MMR-deficiency and MSI-high status are currently the biomarkers employed to select anti-PD-1 treatment [77]. In addition, other agents targeting specific mutations or pathways frequently altered in EC are under evaluation in different clinical trials. These include the PIK3CA pathway (PIK3CA or mTOR inhibitors), *PTEN* mutations (mTOR inhibitors or PARP inhibitors), HRD (PARP inhibitors), *ARID1* mutations (EZH2 inhibitors, or PARP inhibitors) and *FGFR2* mutations (FGFR inhibitors) [19]. Taking into account the therapeutic scenario of advanced EC, the identification of biomarkers to assess disease response and predict recurrence would be critical to create individualized and effective treatment plans to provide the patients with the best long-term survival.

HE4 levels have been proposed as a useful tool for surveillance of cancer recurrence, but their value in monitoring the tumour burden in advanced disease is unknown [78]. Apart from serum markers, one of the most evident strategies to reach this individualized management is based on ctDNA monitoring, as it provides the opportunity to characterize the tumor’s genetic and epigenetic profile at recurrence and during the disease evolution in response to the therapy. We reported a group of endometrial tumours in patients that relapsed after surgery and how the assessment by ddPCR of specific mutations identified in *PIK3CA*, *BRAF* and *CTNNB1* genes in the uterine aspirates may serve as an indicator of the tumour burden [44]. In the case with altered *PIK3CA*, we generated a PDX model which resembled the original tumour characteristics and successfully tested the activity of BYL719 (PI3KCA inhibitor) in these preclinical models, showing the clinical potential of ctDNA monitoring to find therapeutic alternatives for advanced disease [44]. Frequent hot mutations in *PIK3CA* and *KRAS* were also analyzed by ddPCR on cfDNA from EC in another recent work, where two cases of advanced EC were monitored based on ctDNA content during chemotherapy. In both patients, ctDNA levels served as an accurate tool to follow-up the therapy response and the tumour burden, improving the information obtained with CA125 measurements [79]. The status of *PIK3CA* was also analyzed in cfDNA from patients with advanced EC in a clinical trial which interrogated the activity response of the PI3K inhibitor Pilaralisib, finding a good concordance with the results obtained in the primary tumour [80]. Moss et al. [59] also longitudinally analyzed the presence of ctDNA and showed its value to anticipate the progression and mirror the response to treatment in advanced EC cases. Of note, they also detected acquired high microsatellite instability (MSI-H) in ctDNA from one patient, whose primary tumour was MSI stable. Although MSI is considered to be an early event during carcinogenesis, the possibility of interrogating its status by means of cfDNA is really attractive as both a follow-up tool and for immunotherapy selection. Actually, our group has obtained promising results in analyzing five microsatellite markers on DNA from uterine aspirates and plasma, including post-surgery longitudinal plasma samples, which served as a mirror of the tumour burden when patients recurred and also in the therapy response (not published data).

Other circulating biomarkers, such as CTCs or cEVs, are even less explored in advanced stages of EC. CTCs longitudinal assessment was reported during the disease evolution in a cohort of 30 patients with advanced EC using an EpCAM dependent strategy. They detected CTCs in 13 patients during treatment, mainly based on chemotherapy, and showed a CTC dynamic in accordance with their clinical and radiological evolution. This study demonstrated the potential of the CTC population as a follow-up tool, although the rate of CTCs positivity was moderate [81].

## 5. Challenges for the Application of Liquid Biopsy as a Monitoring Tool in EC

The application of minimally invasive follow-up strategies in the context of EC constitutes a key element for advancing in the application of personalized medicine in this tumour. As we summarized in this review, there is scientific evidence about the potential of different circulating biomarkers for improving the surveillance and disease monitoring in the context of non-advanced and advanced tumours (Table 1). Most of the recent studies developed in patients with EC are focused on ctDNA analysis, which is the biomarker with the highest possibility of reaching clinical practice in the near future. Among the different strategies to analyze cfDNA, the identification of specific tumour methylation signatures has emerged in the last years as an accurate option for the early detection and monitoring of different tumours [82,83,84]. Of note, the methylation of genes such as *TBX2*, *CHST11, NID2, ZNF154, PCDH, TNXB* and *DPP6* in tissue samples from patients with EC has been associated with a poor prognosis in low-risk and high-risk EC patients [85,86]. Besides, *HOXA11* methylation status has shown value to predict recurrence in stage I and II EC [87]. Therefore, the characterization of these methylated signatures, among others, in cfDNA from fluid samples may have an important value when looking for possible circulating biomarkers in EC and represents a promising line of research.

In the same line, the study of cEVs is emerging as a valuable source of protein, and genetic biomarkers with increasing data supporting their potential for the diagnosis of gynecologic tumours including EC, but with important work to do for clarifying their potential to anticipate tumour recurrence, guide therapy selection or identify the appearance of resistance to treatment prior to image evaluation. The identification of tumour-specific EVs biomarkers and the development of new isolation technologies suitable for their clinical implementation, such as EXoGAG, successfully applied to characterize ANXAII in plasma from EC patients [88], are important factors to translate cEVs analysis into the cancer patients’ management.

On the other hand, the application of CTCs monitoring and characterization with clinical intention in patients with EC has as a principal limitation the low rate of CTCs that normally characterize this tumour. This fact does not impact on the biological significance of the CTCs dynamics but implies that the majority of non-advanced tumours will present no CTCs during the clinical course of the disease. Therefore, the future of CTCs in EC will probably continue to be associated more with the translational research that tries to understand the mechanisms behind tumour dissemination.

In the context of localized disease in young women, fertility-sparing treatment is of increasing interest in young women. Thus, Casadio et al. showed the convenience of conservative treatment to preserve the fertility opportunities of young patients [89]. The selection of female candidates for this fertility-sparing treatment could be improved by means of liquid biopsy-based approaches such as cfDNA determination.

In the era of immunotherapy, which is also changing the therapeutic options for treatment of re-current/advanced endometrial tumours, oncologists need to complement the current selection biomarker, MSI or MMR deficiency analyzed in tissue samples, with other accurate biomarkers that can improve the patients’ stratification for receiving this therapy and the response assessment. For this purpose, cfDNA- and cEVs-based studies will be of great relevance in the near future due to the important information about the tumour and its microenvironment that can be obtained through their analysis. In this sense, the characterization of circulating immune cells would be also relevant for the evaluation checkpoints-inhibition response.

In addition, the development of robust preclinical models able to recapitulate the tumour molecular characteristics taking into account the heterogeneity of EC is essential to validate the new targeted therapies in a personalized approach. Different pre-clinical models have been developed for drug screening in EC, ranging from the conventional 2D cultures to the patient--derived xenograft (PDXs). However, 2D models are not representative enough of intra-tumour heterogeneity and fail to mimic specific interactions between the tumour and stroma [90]. Notably, patient-derived organoids (PDO) are able to preserve part of the tumour architecture and the molecular heterogeneity, providing more accurate patient-specific responses than simplistic cell line models while also avoiding the use of expensive and time-consuming murine models. Importantly, PDOs from EC have been generated recently and used to test cytotoxic agents, demonstrating their potential as a valuable preclinical tool [91,92,93]. These PDO models are important tools to better capture the tumour microenvironment interactions that condition therapy response, as happens in the case of immunotherapy [94]. For example, Bi et al. have generated PDOs from ovarian and endometrial tumors and demonstrated the capacity of these models to predict the response to therapy [95]. Of note, the analysis of longitudinal liquid biopsies would be relevant for guiding the therapy selection screening in EC PDOs according to the real time genomic information and also as a source of cellular material to generate the organoids and represent the molecular tumour diversity.

**Table 1 cancers-14-01405-t001:** Summary of studies characterizing circulating biomarkers to monitor EC.

Biomarker	Stage	Clinical Significance	Type of Sample	Cohort	Technology	References
HE4 and CA125	Early stages	Prognosis and recurrence monitoring	Serum	174	Enzyme immunoassay	[48]
cfDNA content	Early and advanced stages	Diagnostic, prognostic, potential application to therapy response	Plasma	*n* = 109; 31 FIGO I, 59 FIGO II, 19 FIGO III	PCR-RFLP	[96]
cfDNA content	Early stages	Prognostic predictor	Serum	*n* = 88	Alu-qPCR	[54]
ctDNA	Early and advanced stages	Prognostic, therapy response	Plasma	*n* = 199; 12 G1, 30 G2, 18 G3	ddPCR *(PIK3CA, KRAS)*	[79]
cfDNA and cfmtDNA	Early and advanced stages	Diagnostic, prognostic, potential application to therapy management	Serum	*n* = 81; 12 G1, 30 G2, 17 G3	RT-qPCR	[52]
ctDNA	Early and advanced stages	Prognostic, therapy response	Tissue, serum	*n* = 44; 17 uterine cancer cases)	WES, ddPCR	[57]
ctDNA	Localized and advanced stages	Disease monitoring	Uterine aspirates, plasma	*n* = 60	ddPCR	[44]
ctDNA	Localized and advanced stages	Disease monitoring	Plasma	*n* = 13	NGS	[59]
ctDNA	Localized stages	Disease monitoring	Plasma	*n* = 9	ddPCR	[60]
miR-135b, miR-205 and miR-30a-3p	Localized stages	Diagnostic and post-surgery monitoring	Plasma	*n* = 24	RT-qPCR	[69]
CTCs	Advanced stages	Therapy response	Whole blood	*n* = 30	CellSearch	[81]

## 6. Conclusions

We are enjoying exciting advances in the field of EC management with the arrival of targeted therapies and immunotherapy, but there is a need to apply more accurate and non-invasive markers to better stratify, follow-up and treat endometrial tumours. The present review shows the great potential of different circulating markers for the tumour burden evaluation, MRD detection, recurrence surveillance and treatment monitoring (Figure 1). Among these markers, the analysis of cfDNA/ctDNA has already been implemented into the clinical routine in other tumour types, but in EC, despite the data supporting its clinical interests, its analysis is mainly performed in research studies. Furthermore, the analysis of genetic and proteomic content of cEVs represents a promising tool to improve the technical sensitivity and also to get information about the relevant processes for tumour dissemination and therapy response. For this reason, many clinical trials are including the analysis of circulating biomarkers in localized and metastatic EC to validate their use in different disease contexts (e.g., NCT04651738, NCT05049538, NCT04456972, NCT03776630).

In this regard, some general questions that should be answered before the implementation of liquid biopsy to monitor EC and other tumours are the optimal timepoints for fluid sample collection, the requirements of the tests employed or the threshold criteria to select or change the treatment based on biomarkers that are normally present in very low concentrations. The inclusion of a longitudinal collection of fluid samples as part of ongoing and future clinical trials will be critical for answering these important questions.

## Figures and Tables

**Figure 1 cancers-14-01405-f001:**
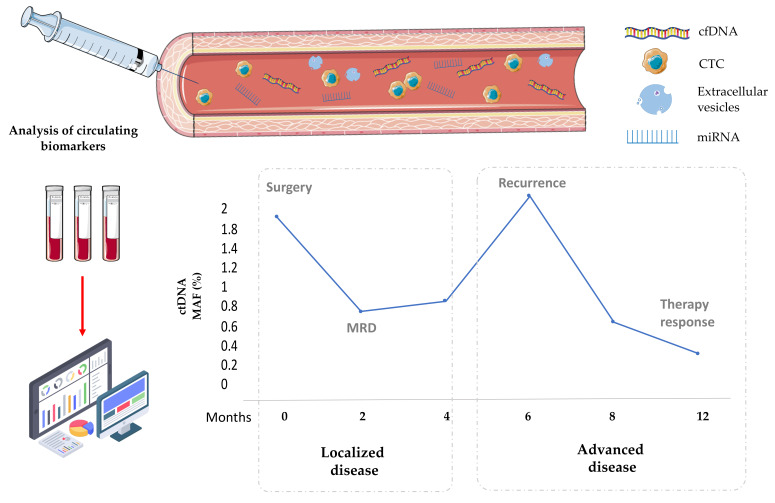
Clinical contexts for the application of liquid biopsy (CTCs, cfDNA, cEVs and miRNAs) to improve the management of patients with both localized and advanced EC. MRD, minimal residual disease. Figures: www.flaticon.com, www.smart.servier.com (accessed on 9 November 2021).

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
