# Peer review of "Liquid Biopsy for Monitoring EC Patients: Towards Personalized Treatment"

_cancers, 2022, doi:10.3390/cancers14061405_

Round 1
Reviewer 1 Report
This is an interesting, well written, review article about the role of circulating biomarkers in endometrial carcinoma (EC), a group of novel, molecular-based and non-invasive, items for monitoring oncologic patients after treatment, that have already demonstrated their efficacy in other malignancies.
Circulating biomarkers have the potential to have a role also in EC diagnosis and follow-up in the future, in the perspective of a tailored, patient-based, approach, which is one of the most exciting challenges in oncology to date.
This review represents an accurate analysis of the state of the art in this field.
I have the following comments to the Authors:
- All manuscripts must contain the required sections: Author Information, Abstract, Keywords, Introduction, Materials & Methods, Results, Conclusions, Figures and Tables with Captions, Funding Information, Author Contributions, Conflict of Interest and other Ethics Statements, as reported in the “Instructions for Authors” section of the journal website: https://www.mdpi.com/journal/cancers/instructions.
In particular, Authors should include the sections “Materials and Methods” and “Results” in the manuscript.
- Authors should check for some mistakes in the text:
- Line 30: there is a repetition (“global perspective”);
- Line 45: after “development”, a comma should be included, and after the comma, an “it” should be added (“it is important”);
- Lines 61-64: Authors should avoid using EEC1, EEC2, EEC3, EEC4 in relation to TCGA groups, since these abbreviations are not used in literature and may be confused. The names of TCGA groups are “POLE ultramutated”, “MSI hypermutated”, “copy-number low” and “copy-number high”, as Authors mentioned in the text.
- Line 124: the abbreviation “cfDNA” is mentioned for the first time in the article without its extended corresponding or meaning, which is described afterwards in the text (line 174);
- Line 150: “an” should replace “and”;
- Line 166: “represents” should replace “represent”;
- Line 178: the subject (“Cicchillitti et al.”) and the verb (“analyzed”) in a sentence should not be separated by a comma;
- Line 213: A period is missing after the reference [59];
- Line 258: An “of” in missing between the words “treatment” and “MMR”;
- Line 317: A reference is required for the sentence “HOXA11 methylation status has shown value to predict recurrence in stage I and II EC”.
- Introduction: In line 81, Authors wrote about studies describing molecular alterations in EC, without citing any. Authors may include in the review a few articles concerning this field (e.g. PMID: 34073635; PMID: 33712277).
- Materials and Methods: Since the article includes a review of the literature, Authors should describe the search strategy, the process of study selection and data extraction, andhow was assessed the risk of bias within studies. Systematic reviews should follow the PRISMA guidelines, as reported in the “Instructions for Authors” section of the journal website: https://www.mdpi.com/journal/cancers/instructions
- In paragraph 3, line 184, Authors cite that cfDNA values have been associated to LVSI, a well-known key prognostic factor for relapse and survival in EC. The association between classic, well-known, clinicopathologic factors in EC and novel molecular ones is a very hot topic in literature to date. Authors may expand the discussion about this including latest evidence in literature.
Author Response
Reviewer 1
This is an interesting, well written, review article about the role of circulating biomarkers in endometrial carcinoma (EC), a group of novel, molecular-based and non-invasive, items for monitoring oncologic patients after treatment, that have already demonstrated their efficacy in other malignancies.
Circulating biomarkers have the potential to have a role also in EC diagnosis and follow-up in the future, in the perspective of a tailored, patient-based, approach, which is one of the most exciting challenges in oncology to date. This review represents an accurate analysis of the state of the art in this field.
-Authors’ comments:
We really appreciate the reviewer comments regarding the scientific interest of our review.
I have the following comments to the Authors:
- All manuscripts must contain the required sections: Author Information, Abstract, Keywords, Introduction, Materials & Methods, Results, Conclusions, Figures and Tables with Captions, Funding Information, Author Contributions, Conflict of Interest and other Ethics Statements, as reported in the “Instructions for Authors” section of the journal website: https://www.mdpi.com/journal/cancers/instructions. In particular, Authors should include the sections “Materials and Methods” and “Results” in the manuscript.
-Authors’ comments:
We appreciate the revisor suggestion since the searching strategy is a key element for understanding the review work. Checking the instructions for authors for preparing reviews we found this recommendation.
“ Reviews: These provide concise and precise updates on the latest progress made in a given area of research. Systematic reviews should follow the PRISMA guidelines. Review articles should be comprehensive and submitted by authors who are in the field. The main text of review papers should be around 4000 words at a minimum and include at least two figures or tables. PRISMA covers systematic reviews and meta-analyses. Authors are recommended to complete the checklist and flow diagram and include it with their submission.”
Since we did not perform a systematic review or meta-analysis the paper structure with material and methods and results is not applicable. In fact, several reviews already published as part of the special issue do not include these sections.
- Authors should check for some mistakes in the text:
- Line 30: there is a repetition (“global perspective”);
- Line 45: after “development”, a comma should be included, and after the comma, an “it” should be added (“it is important”);
- Lines 61-64: Authors should avoid using EEC1, EEC2, EEC3, EEC4 in relation to TCGA groups, since these abbreviations are not used in literature and may be confused. The names of TCGA groups are “POLE ultramutated”, “MSI hypermutated”, “copy-number low” and “copy-number high”, as Authors mentioned in the text.
- Line 124: the abbreviation “cfDNA” is mentioned for the first time in the article without its extended corresponding or meaning, which is described afterwards in the text (line 174);
- Line 150: “an” should replace “and”;
- Line 166: “represents” should replace “represent”;
- Line 178: the subject (“Cicchillitti et al.”) and the verb (“analyzed”) in a sentence should not be separated by a comma;
- Line 213: A period is missing after the reference [59];
- Line 258: An “of” in missing between the words “treatment” and “MMR”;
- Line 317: A reference is required for the sentence “HOXA11 methylation status has shown value to predict recurrence in stage I and II EC”.
-Authors’ comments:
All the corrections and suggestions have been addressed in the new version of the manuscript.
- Introduction: In line 81, Authors wrote about studies describing molecular alterations in EC, without citing any. Authors may include in the review a few articles concerning this field (e.g. PMID: 34073635; PMID: 33712277).
-Authors’ comments:
These two references have been included in the new version of the introduction.
- Materials and Methods: Since the article includes a review of the literature, Authors should describe the search strategy, the process of study selection and data extraction, and how was assessed the risk of bias within studies. Systematic reviews should follow the PRISMA guidelines, as reported in the “Instructions for Authors” section of the journal website: https://www.mdpi.com/journal/cancers/instructions.
-Authors’ comments:
As previously commented, our work is not a systematic review or meta-analysis, therefore, the structure with material and methods and results is not mandatory.
- In paragraph 3, line 184, Authors cite that cfDNA values have been associated to LVSI, a well-known key prognostic factor for relapse and survival in EC. The association between classic, well-known, clinicopathologic factors in EC and novel molecular ones is a very hot topic in literature to date. Authors may expand the discussion about this including latest evidence in literature.
-Authors’ comments:
We appreciate the reviewer's comment. We did not show so many results about the association of circulating biomarkers and risk factors because the focus of the work is the application of liquid biopsies for disease monitoring. However, following the reviewer's suggestion, we have commented in more detail the results about the association between cfDNA and risk of recurrence factors including three references: Gressel et al., Journal of Translational Medicine, 2020; Casas-Arozamena et al., Journal of Clinical Medicine, 2020; Cicchillitti et al., Oncotarget 2017.

Reviewer 2 Report
In this review paper, the authors reviewed the applications of liquid biopsy on evaluating and monitoring advanced endometrial cancer. The authors introduced the progress of several circulating markers as non- invasive tools to monitor endometrial tumor, including ctDNA, CTC and cEVs. This review can help the readers to understand the current status of liquid biopsy application in endometrial cancer. The paper is well written. Just a few points need to be clarified:
- In abstract, line 30, there are two "global perspective" in a sentence;
- line 168, could the authors add more details to HE4 and CA125? How these markers been used in clinic?
- Line 204, what is EpCAM mean?
- In this paper, the authors used "nowadays" for many times, there maybe some words can replace it
- For the PDO models in endometrial cancer, Bi et al (PMID:34200645) have successfully built a platform for endometrial cancer organoid culture and showed that PDO can predict patient response to chemotherapy.
- The authors mentioned the clinical trials ongoing or finished by using liquid biopsy. In my opinion, the trial number should be listed.
Author Response
Reviewer: 2
In this review paper, the authors reviewed the applications of liquid biopsy on evaluating and monitoring advanced endometrial cancer. The authors introduced the progress of several circulating markers as non- invasive tools to monitor endometrial tumor, including ctDNA, CTC and cEVs. This review can help the readers to understand the current status of liquid biopsy application in endometrial cancer. The paper is well written. Just a few points need to be clarified:
- 1. In abstract, line 30, there are two "global perspective" in a sentence;
- line 168, could the authors add more details to HE4 and CA125? How these markers been used in clinic?
- Line 204, what is EpCAM mean?
- In this paper, the authors used "nowadays" for many times, there maybe some words can replace it
- For the PDO models in endometrial cancer, Bi et al (PMID:34200645) have successfully built a platform for endometrial cancer organoid culture and showed that PDO can predict patient response to chemotherapy.
- The authors mentioned the clinical trials ongoing or finished by using liquid biopsy. In my opinion, the trial number should be listed.
-Authors’ comments:
First of all, we would like to thank the reviewer for all the comments and suggestions made about our manuscript. Taking them into account, we have corrected line 30, added the EpCAM meaning, reduced the use of the term “nowadays” and included the reference (NCT04651738, NCT05049538, NCT04456972, NCT03776630) of some ongoing clinical trials which include the analysis of circulating biomarkers (line 564). Moreover, we have included a more detailed comment about the current clinical use of HE4 and CA125 markers in the context of EC (lines 226 to 233). Finally, we also included the reference regarding the PDOs models generated by Bi et al., Cancers 2021 (line 548) as suggested by the reviewer.

Reviewer 3 Report
I had the pleasure to review this well written manuscript for Cancers. This is an interesting review about liquid biopsy for monitoring women affected by endometrial cancer. The topic is attractive and the focus on molecular characterization of endometrial cancer is valuable. However, additional clarifications are needed in order to better interpret the findings of this study.
First, an extensive correction of language is mandatory. Several mistakes about spell and syntax are present in all the manuscript.
Introduction seems to be too long with several unnecessary sentences for this section. In my opinion you could remove them (e.g. Line 52-57; Line 67-80).
Myometrial invasion is considered one of the most important factors for the risk of recurrence and lymph nodes metastasis. However, several studies started to evaluate outcomes for young women affected by endometrial cancer and treated by fertility sparing approach (Casadio et al. Fertility Sparing Treatment of Endometrial Cancer with and without Initial Infiltration of Myometrium: A Single Center Experience. Cancers (Basel). 2020 Nov 29;12(12):3571. doi: 10.3390/cancers12123571. Please discuss it.
Figure 1 does not add anything to the manuscript. The route of the curved line is hardly understandable.
Author Response
Reviewer: 3:
I had the pleasure to review this well written manuscript for Cancers. This is an interesting review about liquid biopsy for monitoring women affected by endometrial cancer. The topic is attractive and the focus on molecular characterization of endometrial cancer is valuable. However, additional clarifications are needed in order to better interpret the findings of this study.
First, an extensive correction of language is mandatory. Several mistakes about spell and syntax are present in all the manuscript.
-Authors’ comments:
We have checked the language style and grammar in order to improve the quality of the manuscript.
Introduction seems to be too long with several unnecessary sentences for this section. In my opinion you could remove them (e.g. Line 52-57; Line 67-80).
-Authors’ comments:
Taking into account the reviewer's suggestion we have reduced the introduction eliminating the part describing risk factors to suffer EC since this information is not relevant for understanding the need for monitoring biomarkers in the context of endometrial cancer.
Myometrial invasion is considered one of the most important factors for the risk of recurrence and lymph nodes metastasis. However, several studies started to evaluate outcomes for young women affected by endometrial cancer and treated by fertility sparing approach (Casadio et al. Fertility Sparing Treatment of Endometrial Cancer with and without Initial Infiltration of Myometrium: A Single Center Experience. Cancers (Basel). 2020 Nov 29;12(12):3571. doi: 10.3390/cancers12123571. Please discuss it.
-Authors’ comments:
We consider the clinical scenario commented by the reviewer of great interest for the application of liquid biopsies. Therefore, we included this point in section 5 (line 519) of the new version.
Figure 1 does not add anything to the manuscript. The route of the curved line is hardly understandable.
-Authors’ comments:
We appreciate the comment. We have updated Figure 1 to make it more informative. In particular, we have included a real graph reflecting ctDNA levels during the different points of the tumor evolution with the goal of making more evident the clinical contexts that may benefit from the use of circulating biomarkers.

Round 2
Reviewer 1 Report
I kindly thank Authors for answering to all my comments.
This review represents an accurate analysis of the the role of circulating biomarkers in endometrial carcinoma (EC) and I am satisfied with the revised version of the manuscript.
Reviewer 3 Report
The authors have adequately improved the manuscript.